# Facial Expressions Track Depressive Symptoms in Old Age

**DOI:** 10.3390/s23167080

**Published:** 2023-08-10

**Authors:** Hairin Kim, Seyul Kwak, So Young Yoo, Eui Chul Lee, Soowon Park, Hyunwoong Ko, Minju Bae, Myogyeong Seo, Gieun Nam, Jun-Young Lee

**Affiliations:** 1Department of Psychiatry, Seoul National University College of Medicine & SMG-SNU Boramae Medical Center, Seoul 07061, Republic of Korea; 2Department of Psychology, Pusan National University, Busan 46241, Republic of Korea; 3Department of Human-Centered Artificial Intelligence, Sangmyung University, Hongjimun 2-Gil 20, Jongno-Gu, Seoul 03016, Republic of Korea; 4Division of Teacher Education, College of General Education for Truth, Sincerity and Love, Kyonggi University, Suwon 16227, Republic of Korea; 5Samsung Advanced Institute for Health Sciences and Technology (SAIHST), Sungkyunkwan University, Samsung Medical Center, Seoul 06355, Republic of Korea; 6Interdisciplinary Program in Cognitive Science, Seoul National University, Seoul 08826, Republic of Korea

**Keywords:** late-life depression, facial expression, digital biomarker, facial muscle, emotion, depressive symptoms

## Abstract

Facial expressions play a crucial role in the diagnosis of mental illnesses characterized by mood changes. The Facial Action Coding System (FACS) is a comprehensive framework that systematically categorizes and captures even subtle changes in facial appearance, enabling the examination of emotional expressions. In this study, we investigated the association between facial expressions and depressive symptoms in a sample of 59 older adults without cognitive impairment. Utilizing the FACS and the Korean version of the Beck Depression Inventory-II, we analyzed both “posed” and “spontaneous” facial expressions across six basic emotions: happiness, sadness, fear, anger, surprise, and disgust. Through principal component analysis, we summarized 17 action units across these emotion conditions. Subsequently, multiple regression analyses were performed to identify specific facial expression features that explain depressive symptoms. Our findings revealed several distinct features of posed and spontaneous facial expressions. Specifically, among older adults with higher depressive symptoms, a posed face exhibited a downward and inward pull at the corner of the mouth, indicative of sadness. In contrast, a spontaneous face displayed raised and narrowed inner brows, which was associated with more severe depressive symptoms in older adults. These findings suggest that facial expressions can provide valuable insights into assessing depressive symptoms in older adults.

## 1. Introduction

Late-life depression is a pressing social issue due to its severe consequences, which include an increased risk of morbidity and suicide [1,2]. Furthermore, depression is linked to reduced physical, cognitive, and social functioning, as well as higher mortality rates [3]. Epidemiological studies revealed that the prevalence of major depression among individuals aged over 75 years ranges from 4.6% to 9.3%, and this percentage rises to 27% for those aged over 85 years [4,5].

In epidemiological studies and clinical settings, self-report measures are commonly employed to evaluate the incidence and severity of depressive symptoms among older adults [6]. It is worth noting that certain typical symptoms of depression, such as cognitive impairment and memory loss, are also observed in older adults who are on the path to developing dementia [7]. However, evaluating depressive symptoms among older adults is not easy because older adults with memory issues may deny experiencing depressive symptoms [8]. Therefore, researchers have emphasized and relied on clinical interviews and observations to capture information such as nonverbal aspects, which are essential for diagnosis [9,10,11].

Analyzing facial expression could aid in diagnosing depression among patients [12]. Previous research has indicated that nonverbal behaviors exhibited in facial regions, particularly changes in facial expressions, are linked to the severity of depressive symptoms [11,13,14]. For example, individuals with depression exhibit reduced facial muscle activity in the brow and chin regions when engaging in tasks involving positive and negative emotional imagery [15]. Additionally, patients with depression display limited movement and attenuated smiles in the zygomatic major muscles [16,17].

The Facial Action Coding System (FACS) is a comprehensive system developed by Paul Ekman and colleagues (1978) to analyze and classify expressions based on the activation and movement of specific facial muscles. The FACS provides a standardized and objective framework for understanding and quantifying facial expressions across different individuals. The FACS consists of a detailed set of anatomical movements called action units (AUs), which represent specific facial muscle actions associated with different emotional expressions. It provides a systematic method for capturing and measuring facial movements, enabling researchers to obtain accurate and consistent data on emotional expressions [18]. The FACS has the ability to capture subtle facial expressions, which may convey important emotional information. These subtle cues might not be easily detected by subjective judgement alone. Moreover, the FACS enhances cross-cultural research by providing a common framework for analyzing across different populations. Since the FACS is based on objective muscle movements rather than subjective interpretations, it allows comparability and generalizability of findings across diverse cultural contexts [19]. Based on its technical advantages, the FACS has been utilized to predict major depressive disorder, post-traumatic stress disorder, and general anxiety disorder [20]. Researchers suggest that emotional expression is visible by the right combination of facial muscles and the FACS provides a system for detecting depression [21].

In the current study, we aimed to investigate older adults’ facial expressions linked to depressive symptoms. Previous studies have revealed a link between depressive symptoms and facial expressions in young adults [15,22]. However, the signs of depression are presented differently in older adults and could be detected using distinctive facial expressions. For instance, relative to younger adults, older adults with depression are less likely to endorse affective symptoms and are more likely to display cognitive changes, somatic symptoms, and loss of interest [23,24]. Therefore, we investigated older adults’ facial expressions using the FACS.

Additionally, we conducted an analysis of both posed and spontaneous facial expressions due to their distinct neural basis, which are documented in the literature. The control of most expressive facial muscles is governed by both cortical and subcortical upper motor neuron circuits. The cortical circuit primarily governs spontaneous facial expressions [25]. These distinct neuroanatomical bases have led to the classification of facial expressions into posed and spontaneous categories. Studies have indicated that posed and spontaneous facial expressions exhibit different dynamic sequences within the same emotion [26]. This means that an intentionally displayed emotional expression, such as a posed expression, and a spontaneous facial expression when people react spontaneously to an emotional stimulus possess different characteristics. Therefore, solely observing either posed or spontaneous facial expressions would provide incomplete understanding of the full range of facial expressions.

## 2. Methods

### 2.1. Participants

Fifty-nine older adults (mean age of 71.98 years, 35 females) without cognitive impairment who were recruited from psychiatric clinics were included in the current analysis (Table 1). To ensure the exclusion of participants with cognitive impairment, we utilized the Clinical Dementia Rating (CDR) [27] and the Mini-Mental State Examination (MMSE) [28]. The participants included in the current study demonstrated a cognitively normal status, as evidenced by the average Mini-Mental State Examination (MMSE) score presented in Table 1. This approach was implemented based on the understanding that depression can be both a risk factor for dementia and a prodrome of dementia [29]. Furthermore, employing these measures aimed to enhance the reliability of the self-reported depression scale.

Additionally, we took into account the participants’ years of education as depression questionnaires can vary in complexity and may be influenced by a respondent’s educational background. It has been suggested that considering educational disparities is important in depression measurement to identify individuals who may encounter difficulties with written questionnaires or may be susceptible to subtle psychometric biases associated with education [30]. Notably, our participants’ average level of education aligns closely with the average education levels of older adults in Korea, whose mean years of education is 8.89 with standard deviation 0.74 [31]. The scores on the Beck Depression Inventory (BDI-II), ranging from 0 to 63 [32], are interpreted as follows: 0–13: normal, 14–19: slightly depressed, 20–28: moderately depressed, and 29–63: severely depressed. In the Korean version of the BDI-II (K-BDI-II), a cutoff score of 17 points has been suggested for a depressive-related disorder [33]. Consequently, the current study included participants exhibiting slight to moderate depressive symptoms based on this criterion.

When older adults visited one of the clinics, individuals who voluntarily expressed their willingness to participate in the study were enrolled after being provided with information about the research. Older adults who did not meet the inclusion criterion of “absence of cognitive impairment” did not take part in this research. All participants provided written informed consent prior to participating in the study. This study was conducted in accordance with the Declaration of Helsinki, and the institutional review board approved the protocol of the SMG-SNU Boramae Medical Center (IRB No. 30-2017-63).

### 2.2. Facial Expression Task

The facial expression task used in a previous study was utilized [19]. This task captured posed and spontaneous facial expressions. A series of photos showing six basic emotions and a neutral facial expression were presented to the participants to capture their posed facial expressions. The participants were then asked to identify the emotions shown in the photos. The following instructions for each emotion were provided: “Face the camera and make a happy (fearful, surprised, angry, sad, disgusted, or neutral) face for 15 s.” To capture spontaneous emotional expressions, the participants were presented with 6 short films lasting approximately 120 s that provided various emotional stimuli, including happiness, fear, surprise, anger, sadness, and disgust. The films including each emotional stimulus were selected based on previous research [19]. Specifically, the fear emotional stimuli were provided by “The Shining (1980)”, the surprise emotional stimuli were provided by “Capricorn One (1978)”, the disgust emotional stimuli were provided by “Pink Flamingos (1972)”, and the sad emotional stimuli were presented by “The Cham (1979)”. The angry emotional stimuli were presented by “The Attorney (2013)”, and the happiness emotional stimuli were presented by “About Time (2012)”. Although the movie clips were selected based on a discussion among the psychologists and psychiatrists involved in this study, considering the previous literature, we admit the possibility that the emotional stimulation might not induce the same emotion in individuals. It is also impossible to assume that any emotional stimulus would cause a single emotion. Therefore, throughout the overall emotional movie watching, we aimed to measure the participants’ spontaneous expressions in the presence of externally given emotional stimuli.

### 2.3. Data Acquisition

The participants’ video recordings of posed facial expressions were captured using a Canon EOS 70D DSLR camera with a 50 mm prime lens, a 720 p resolution, and a 60 fps frame rate. The camera was positioned on a fixed stand approximately 120–140 cm above the floor to correctly capture the entire face of each participant. The posed facial expressions were recorded for 15 s after the participants were instructed to imitate a previously recognized emotional face. For each frame of the recorded videos, the presence and intensity were estimated using OpenFace 2.0, an open-source toolkit for facial behavior analysis that consists of four pipelines: (1) facial landmark detection and tracking, (2) head pose estimation, (3) eye gaze estimation, and (4) facial expression recognition. OpenFace 2.0 recognizes facial expressions by detecting action unit (AU) intensity and presence according to the FACS [34]. Without using all the AUs listed in the FACS, OpenFace 2.0 offers a subset of the following 17 AUs through cross-dataset learning: 01, 02, 04, 05, 06, 07, 09, 10, 12, 14, 15, 17, 20, 23, 25, 26, and 45. The occurrence and intensity of AUs are estimated using machine learning algorithms. The AU estimation and analysis methods are described in more detail in the literature [35]. In the current study, AU intensities were used to derive individual emotional facial expression measures, and six basic emotions were created according to the emotional FACS (EMFACS). The EMFACS is based on the FACS and has been proven to have significant reliability for assessing human facial movements [36,37]. The highest intensity for each AU was calculated as the maximum score across all video frames, as established in a previous study [38].

### 2.4. Korean Version of the Beck Depression Inventory-II

To measure depressive symptoms, this study utilized the Korean version of the Beck Depression Inventory-II (K-BDI-II) [39]. The K-BDI-II consists of 21 questions evaluating the severity of depression, with scores ranging from 0 to 63 [32].

### 2.5. Statistical Analysis

We performed principal component analysis (PCA) to overcome the high dimensionality and multicollinearity of facial AUs. The parallel analysis determined the number of principal components as 8. Therefore, we summarized the 17 spontaneous AUs of the 6 basic emotions into 8 principal components. Also, we summarized the 17 posed AUs of the basic emotions and the neutral faces made by the participants. In addition to this, we generated principal component scores using a single emotion of the six basic emotions.

To identify the characteristics of specific emotions that predict depressive symptoms, we also calculated the principal component scores for the neutral faces and basic emotions, including happiness, sadness, fear, anger, surprise, and disgust. Parallel analysis was performed to determine the optimal number of components for further analysis. This analysis compares the degree of eigenvalues of the observed data with that of a random data matrix of the same size as the original. Herein, a random data matrix was generated with 50 iterations. The components with eigenvalues higher than those of the randomly generated data were considered meaningful units of the principal components. The parallel analysis was conducted using the psych package [40]. The current measures of the posed action units (PAUs) and spontaneous action units (SAUs) identified eight principal components as being optimal.

After dimension reduction, multiple regression analysis was performed to identify the principal component scores relevant to the total K-BDI-II score. The multiple regression analysis included age, education, sex, and cognitive function as covariates of no interest. For the principal components scores generated from a single basic emotion condition, we ran a stepwise multiple regression analysis to determine whether the principal component scores improved the model fit for predicting depressive symptoms. For instance, we added either the principal scores for the posed action units or the principal scores for the spontaneous action units into the baseline model, which included demographic data (age, sex, education, and MMSE score). The relative contribution of the AUs that showed a significant association with depressive symptoms was visualized using the Py-Feat toolbox [41].

## 3. Results

The proportions of variance explained by the suggested number of principal components in the posed and spontaneous conditions were 63.1% and 65.2%, respectively. In the posed condition, the first three principal components explained 21.9%, 9.6%, and 8% of the variance. In the spontaneous condition, the first three principal components explained 24.3%, 8.6%, and 8.3% of the variance. Table 2 shows the association between depressive symptoms and posed and spontaneous facial expressions. In the model of posed action units, PC3 significantly predicted depressive symptoms, as measured by the total K-BDI-II scores, among the principal component scores from the posed facial AUs. In the model of spontaneous action units, Spontaneous-PC5 and Spontaneous-PC7 were significantly associated with the total K-BDI-II scores.

In order to elucidate the relationship between the overall posed and spontaneous facial expressions and the Beck Depression Inventory-II (BDI-II) scores, we calculated the R-squared value. The R-squared value signifies the proportion of variance in the dependent variable that can explained by the independent variables in a regression model. We incorporated either the posed-action-unit variable or the spontaneous-action-unit variable into the baseline regression model, which solely included demographic data (age, sex, education, and MMSE score). Table 3 presents the percentage of variance explained by the collective action unit information for both spontaneous and posed facial expressions (Table 3, emotion condition: all emotions). As the current study used eight principal components for the PCA analysis, the all-emotion condition comprised eight principal components. Additionally, the table includes the R-squared change value, which demonstrates that 20.7% of the variance is explained by incorporating information on posed facial expressions into the model predicting the BDI-II scores. Similarly, the inclusion of information on spontaneous facial expressions in the model resulted in a 21.4% increase in the explained variance for the BDI-II score prediction.

Furthermore, we computed the R-squared change value to assess the impact of adding the action unit information for each of the six basic emotions to the baseline model consisting of demographic variables (age, sex, education, and MMSE score). Among the six basic emotions, fear and disgust displayed in the posed facial expressions demonstrated a relatively substantial improvement in the explained variance of the model (with an R-squared change of 0.067 being attributed to posed action units in the fear and disgust conditions). Concerning the spontaneous facial expressions, fear exhibited the highest improvement among the six basic emotions (with an R-squared change of 0.033 being attributed to spontaneous action units in the fear condition). However, for both posed and spontaneous facial expressions, the specific emotion conditions resulted in a smaller R-squared change due to the principal components compared to the model that encompassed all of the emotions.

Table 4 presents the action units (AUs) that significantly contributed to the principal components, which in turn predicted depressive symptom scores in the regression model. Additionally, we provided information on the corresponding facial muscles associated with these AUs. In the posed facial expressions, participants with higher depressive symptom scores exhibited an increased expression in facial muscles involved in blinking, depressed, and tight lip movements. Conversely, muscles related to lowering the eyebrows, which are involved in various emotional reactions such as fear, anger, and disgust, were weaker in participants with higher depressive symptom scores [42]. The factor loadings of the posed AUs onto the principal components are visually represented in Figure 1a, with a darker color indicating larger scores for both negative and positive factor loadings. In terms of spontaneous facial expressions, participants with higher depressive symptom scores displayed more prominent facial muscle movements related to raised inner brow, blinking, and wrinkled nose. However, movements of a dimpled cheek were minimized in these participants. Figure 1b,c depict the facial mapping of factor loadings for each action unit in the spontaneous expressions.

## 4. Discussion

In this study, we examined the facial expressions of older adults without cognitive impairment using two different measurements: posed and spontaneous facial expressions. Our findings revealed distinct features in the facial expressions of older adults with more severe depressive symptoms compared to those with milder or no symptoms. Specifically, we observed that AU15, which involves the downward and inward movement of the corner of the mouth, was particularly prominent among the posed action units exhibited by older adults with more severe depressive symptoms. This movement is commonly associated with sadness [43].

In addition, we found a pattern differentiating older adults with depression from others through the spontaneous facial expressions shown as they watched a series of videos including emotional stimuli. As depressive symptoms increased, the movements in the lower part of the face were attenuated while the upper face was significantly involved during spontaneous facial expressions. In particular, the raising and narrowing of the inner corners of the brows (AU1 + AU4), which have been observed among older adults with depression, are common signs of sadness [44]. AU12, which is less likely to be observed among older adults with depression, is the main component of a smiling face with an open mouth [43].

A separate neural basis might explain the discrepancy between the posed and spontaneous facial expressions. Posed and spontaneous facial behaviors are mediated by separate motor pathways, namely, the pyramidal and extrapyramidal motor tracks, respectively [45]. Researchers have suggested that spontaneous (genuine) expressions occur as part of an emotional experience. Posed expressions are not coupled with a corresponding emotion and occur as a means of faking, masking, or suppressing emotional experiences [44]. Thus, posed expressions provide limited information regarding the actual affective state of a person. Posed facial expressions are also known as deliberate facial expressions, which are voluntary and intentional facial movements. Researchers have shifted their focus from posed facial expressions to spontaneous facial expressions, which are more likely to be observed in realistic situations [46].

To identify a specific emotional condition that maximizes the facial features associated with depressive symptoms, we analyzed six basic emotions in this study. The results indicated that adding facial AU data from the fear and disgust conditions to the linear regression model predicting depressive symptoms generated the largest R-squared change among the six basic emotions. This pattern was observed in the posed face data. Although the fear condition showed the highest R-squared change for spontaneous facial expressions, the R-squared change was smaller than that from the posed facial expressions in the fear condition. In other words, when older adults were asked to mimic fearful and disgusted emotional faces, the features associated with depressive symptoms were maximized.

The findings of this study highlight the importance of understanding the facial expression system in detecting affective disorders through nonverbal communication. This study represents a significant advancement by directly assessing nonverbal behaviors based on facial expressions. While previous research has primarily focused on facial expression recognition ability, this study takes a step forward by directly capturing individuals’ facial expressions [47,48]. Other studies have explored participants’ facial expressions, but they have also relied on human raters, which may introduce biases [49]. In contrast, our study utilizes a standardized analysis method to examine nonverbal features of depressive symptoms, providing a more objective approach. In addition to this, our findings support a separate facial expression system for posed and spontaneous facial expressions [26].

This study has several limitations. We recruited participants who were not diagnosed with a major depressive disorder in this study. Even if we revealed an association between depressive symptoms and facial expressions, the generalizability of the current findings could be limited because we did not include a clinical population with a major depressive disorder. In addition, we focused on facial expressions shown toward general emotional stimuli, rather than a specific emotional condition, to identify older adults suffering from depressive symptoms using related features of facial expressions. We examined all combinations of six basic emotions and 17 AUs to select features that predict depressive symptoms. The fact that we did not investigate facial expressions with a focus on a specific emotion, such as sadness or happiness, could be noted as a limitation of the current study.

## Figures and Tables

**Figure 1 sensors-23-07080-f001:**
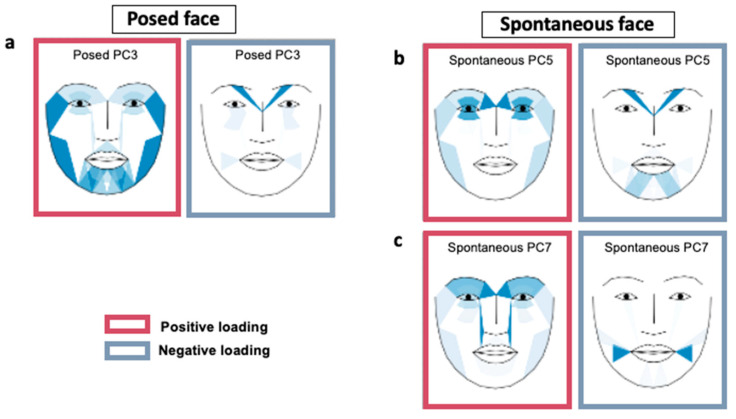
Facial heat map presenting the principal component scores predicting K-BDI-II scores (**a**–**c**). The principal component score of each action unit is presented in Table 4. A darker color indicates a larger absolute value of the principal component score.

**Table 1 sensors-23-07080-t001:** Demographic characteristics of participants.

	Mean (SD)/Frequency (Proportion)
Age	71.98 (6.11)
Sex (male:female)	24:35
Education (years)	9.82 (4.03)
MMSE	26.17 (3.12)
K-BDI-II	18.37 (12.71)

MMSE: Mini-Mental State Examination; K-BDI-II: Korean version of the Beck Depression Inventory-II.

**Table 2 sensors-23-07080-t002:** Multiple regression results explaining K-BDI-II scores.

	Posed AU			Spontaneous AU	
	B(SE)	*p*-Value		B(SE)	*p*-Value
Age	−0.50 (0.28)	0.081	Age	−0.37 (0.27)	0.179
Sex	−3.05 (4.21)	0.472	Sex	−3.50 (4.12)	0.400
Education	−0.17 (0.44)	0.709	Education	−0.22 (0.44)	0.619
MMSE	−1.01 (0.6)	0.100	MMSE	−1.06 (0.59)	0.077
P-PC1	0.28 (0.52)	0.585	S-PC1	−0.16 (0.42)	0.706
P-PC2	−1.06 (0.75)	0.164	S-PC2	−0.39 (0.73)	0.594
**P-PC3**	**2.09 (0.87)**	**0.021**	S-PC3	0.57 (0.79)	0.474
P-PC4	−0.04 (0.95)	0.969	S-PC4	0.69 (0.87)	0.436
P-PC5	1.64 (1.07)	0.134	**S-PC5**	**2.73 (0.94)**	**0.006**
P-PC6	2.06 (1.06)	0.057	S-PC6	−0.71 (0.98)	0.474
P-PC7	0.80 (1.13)	0.483	**S-PC7**	**2.34 (1.02)**	**0.027**
P-PC8	−0.65 (1.39)	0.641	S-PC8	0.24 (1.19)	0.842

AU: action unit; MMSE: Mini-Mental State Exam; P-PC: posed principal component; S-PC: spontaneous principal component.

**Table 3 sensors-23-07080-t003:** R-squared changes after adding the principal component scores in the linear regression model predicting K-BDI-II scores.

Emotion Condition	Posed		Spontaneous	
	Number of PCs	R-Squared Change	Number of PCs	R-Squared Change
All emotions	8	0.207	8	0.214
Neutral	2	0.038	NA	NA
Fear	2	0.067	2	0.033
Disgust	2	0.067	2	0.016
Anger	3	0.039	2	0.014
Sad	1	0.038	1	0.003
Surprise	1	0.030	1	0.028
Happy	1	0.027	2	0.000

All emotions indicate the principal component (PC) scores generated by the sum of the six basic emotions.

**Table 4 sensors-23-07080-t004:** Principal component score of each action unit predicting BDI-II scores.

AU	Description	Location	PAU PC3	SAU PC5	SAU PC7
AU01	Raised inner brow	Upper	0.036	0.156	0.222
AU02	Raised outer brow	Upper	0.073	0.071	0.171
AU04	Lowered brow	Upper	−0.381	−0.205	−0.011
AU06	Raised cheek	Upper	−0.128	0.019	0.071
AU17	Raised chin	Upper	0.097	−0.041	−0.077
AU45	Blinking	Upper	0.610	0.281	0.317
AU05	Raised upper lid	Lower	−0.011	−0.039	0.072
AU07	Tight lid	Lower	0.081	0.133	0.117
AU09	Wrinkled nose	Lower	0.067	0.032	0.225
AU10	Raised upper lip	Lower	−0.045	0.008	−0.057
AU12	Pulled lip corner	Lower	−0.010	0.014	−0.118
AU14	Dimpled	Lower	−0.143	−0.053	−0.371
AU15	Depressed lip corner	Lower	0.103	−0.080	0.007
AU20	Stretched lip	Lower	0.081	−0.040	0.051
AU23	Tight lip	Lower	0.151	−0.133	−0.005
AU25	Parted lips	Lower	0.077	−0.006	0.088
AU26	Dropped jaw	Lower	0.150	0.069	0.074

## Data Availability

Data will be available upon request.

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
