# Peer review of "Facial Expressions Track Depressive Symptoms in Old Age"

_sensors, 2023, doi:10.3390/s23167080_

Round 1

Reviewer 1 Report

The paper examines the relationship between facial expressions and depressive symptoms among older adults without cognitive impairment by using FACS and the Korean version of the Beck Depression Inventory-II scale and by encoding posed and spontaneous facial expressions across six basic emotions.

The readability of the paper could be improved by adding an outline at the end of the introduction section.

In addition, the technical contribution of this paper is not clear. The author should state the technical novelty clearly. In particular, a related work section needs to be added by discussing studies on the relationship between facial expressions and depressive symptoms. Some examples of works to discuss are:

·         Ruihua M, Hua G, Meng Z, Nan C, Panqi L, Sijia L, Jing S, Yunlong T, Shuping T, Fude Y, Li T and Zhiren W (2021) The Relationship Between Facial Expression and Cognitive Function in Patients With Depression. Front. Psychol. 12:648346. doi: 10.3389/fpsyg.2021.648346

·         Park, S.; Lee, K.; Lim, J.-A.; Ko, H.; Kim, T.; Lee, J.-I.; Kim, H.; Han, S.-J.; Kim, J.-S.; Park, S.; Lee, J.-Y.; Lee, E.C. Differences in Facial Expressions between Spontaneous and Posed Smiles: Automated Method by Action Units and Three-Dimensional Facial Landmarks. Sensors 202020, 1199. https://doi.org/10.3390/s20041199

·         Gang Fu, Yanhong Yu, Jiayu Ye, Yunshao Zheng, Wentao Li, Ning Cui, Qingxiang Wang, A method for diagnosing depression: Facial expression mimicry is evaluated by facial expression recognition, Journal of Affective Disorders, Volume 323, 2023, Pages 809-818, ISSN 0165-0327, https://doi.org/10.1016/j.jad.2022.12.029

·         Maximiano-Barreto MA, Bomfim AJL, Borges MM, de Moura AB, Luchesi BM, Chagas MHN. Recognition of Facial Expressions of Emotion and Depressive Symptoms among Caregivers with Different Levels of Empathy. Clin Gerontol. 2022 Oct-Dec;45(5):1245-1252. doi: 10.1080/07317115.2021.1937426. Epub 2021 Jul 5. PMID: 34219607.

The authors should provide a discussion characterizing each work, its benefits and drawbacks, and a comparison among the related works to justify the necessity of the study proposed in the paper. Without any comparison, it is not possible to evaluate the originality degree of the work.

Please state clearly and precisely in the paper what makes original this work.

Author Response

We appreciate your thoughtful comments. Please see the attachment.

Reviewer 2 Report

I would like to express my gratitude for the opportunity to read this research. The recommendations in this document are intended to help you improve your work. Here are a few small points to consider:

I would suggest authors adding information about specific research question and why this study important to do in introduction?

I would suggest authors  to explain about the meaning of mean education, MMSE and K-BDI-II and its correlation with the selection of  your participant?

Please read carefully   Line 176, please explain about  correlation between posed face and spontaneous face with KBDI-II?

What the correlation between  all emotion in table 3 with depressive symptoms?

Author Response

(The authors gave the same response as above.)

Reviewer 3 Report

The paper reports a study on the relationships between facial expression features and depression, through the use of facial data collected within the parameters (action units) of the Facial Action Coding System (FACS). I find the study well-designed, the data complex and well analyzed, and the results valuable for the topic of the facial markers of depression. There are only a few issues that I consider to require further explanation, as listed below.

The FACS and its previous use in exploring psychopathology should be also presented in the Introduction.

The participants recruitment procedure should be presented.

The reasoning behind including the MMSE scores as distinct predictor of the K-BDI-II score should be presented.

The information in Table 4 needs to be also commented in the Results section.

Line 274 “Nevertheless, we maximized interpretability by presenting the factor loading score of each AU and explaining the associated emotion of the significant AU revealed in a previous study.” – this needs to be further explained; which study? And how did it guide (if applicable) the selection of the AUs related to each emotion?

Author Response

(The authors gave the same response as above.)

Reviewer 4 Report

Following observations are made  on  the  paper "Facial expressions track depressive symptoms in old age"  

(i) There is no novelty in the  proposed work

(ii) Abstract  is not proper

(iiI) Lacks in  connection between depression measurement with  different facial  expression

(iV)  Dataset size is very much less

(v) No  evidence ethical report from the hospital is enclosed 

(vi) There is lack of quantitative parameters used in  result

(vii)  Old references, few references are not properly listed

Author Response

(The authors gave the same response as above.)

Round 2

Reviewer 1 Report

The authors addressed my comments.

Author Response

Thank you

Reviewer 2 Report

Dear author,

Thank you for your hard work on revising this manuscript. Hope to get good news from the editor soon.

Author Response

Thank you

Reviewer 3 Report

In my view, the revised manuscript can be accepted for publication.

Author Response

Thank you